# Bone Morphogenetic Protein 2 Enhances Porcine Beige Adipogenesis via AKT/mTOR and MAPK Signaling Pathways

**DOI:** 10.3390/ijms25073915

**Published:** 2024-03-31

**Authors:** Jiali Liu, Yao Jiang, Chuanhe Chen, Lilan Zhang, Jiahao Wang, Chunhuai Yang, Tianwen Wu, Shulin Yang, Cong Tao, Yanfang Wang

**Affiliations:** 1State Key Laboratory of Animal Biotech Breeding, Institute of Animal Science, Chinese Academy of Agricultural Sciences (CAAS), Beijing 100193, China; adai1013@126.com (J.L.); chenchuanhe2022@126.com (C.C.); zhanglilancaas@163.com (L.Z.); wangjiahaobio@163.com (J.W.); yangch023@foxmail.com (C.Y.); wutianwen@caas.cn (T.W.); yangshulin@caas.cn (S.Y.); 2National Animal Husbandry Service, Beijing 100125, China; jiangyao133996@126.com

**Keywords:** pig, BMP2, beige adipocytes, adipogenesis, PPARγ

## Abstract

Bone morphogenetic protein 2 (BMP2) has been reported to regulate adipogenesis, but its role in porcine beige adipocyte formation remains unclear. Our data reveal that *BMP2* is significantly induced at the early stages of porcine beige adipocyte differentiation. Additionally, supplementing rhBMP2 during the early stages, but not the late stages of differentiation, significantly enhances porcine SVF adipogenesis, thermogenesis, and proliferation. Furthermore, compared to the empty plasmid-transfected-SVFs, *BMP2*-overexpressed SVFs had the enhanced lipid accumulation and thermogenesis, while knockdown of *BMP2* in SVFs exhibited the opposite effect. The RNA-seq of the above three types of cells revealed the enrichment of the annotation of thermogenesis, brown cell differentiation, etc. In addition, the analysis also highlights the significant enrichment of cell adhesion, the MAPK cascade, and PPARγ signaling. Mechanistically, *BMP2* positively regulates the adipogenic and thermogenic capacities of porcine beige adipocytes by activating PPARγ expression through AKT/mTOR and MAPK signaling pathways.

## 1. Introduction

Beyond its critical role in energy storage, adipose tissue is now increasingly being recognized as a vital, complex endocrine organ-secreting signaling molecule and hormone that regulates metabolic homeostasis [1]. As the prevalence of human obesity dramatically increases over the world, adipose biology becomes a hotspot in the research field of metabolism. In the swine industry, fat deposition is very important, as subcutaneous and visceral fats affect the lean meat percentage and feed efficiency, while the intramuscular fat (IMF) content determines meat quality and flavor [2]. However, an understanding of fat deposition in pigs at the molecular and cellular levels is limited.

With the rapid development of biotechnologies, our understanding of adipose tissues and adipocytes is greatly expanding. Most placental mammals boast two principal categories of adipose tissues, white and brown [3,4,5], and they are mainly composed of white and brown adipocytes, respectively. White adipocytes, characterized by their round and large unilocular lipid droplets [6], are the prevalent adipocyte type in the body and accumulate fat as an energy source [7]. In contrast, brown adipocytes distinguish themselves with a profusion of mitochondria and multilocular lipid droplets. The third adipocyte, beige adipocytes, also known as brite (brown in white) adipocytes [8], were discovered in white adipose tissues upon stimulation, such as cold exposure, and showed an intermediate phenotype between those of white and brown adipocytes. Except for cellular differences, three types of adipocytes displayed distinct molecular features, as evidenced by the different molecular markers TCF21 for white adipocytes [9], UCP1 for brown adipocytes, and CD137 for beige adipocytes [10,11]. The role of beige adipocytes is significant in the regulation of energy balance and metabolism. They are particularly important in response to cold environments, where they help to maintain body temperature by burning stored fat to produce heat. Further, these thermogenic cells, when active, have a high rate of nutrient consumption and energy expenditure; their existence in adult humans not only correlates with improved metabolic profiles but has stimulated interest in targeting them therapeutically to fight obesity, improve insulin sensitivity, reduce inflammation, and improve glycemic control [12,13,14].

Accumulating evidence revealed that neither brown adipose tissues nor functional UCP1 exist in pigs, which makes them susceptible to cold and prone to fat deposition and results in neonatal mortality and decreased production efficiency. In our previous study, a CRISPR/Cas9-mediated homologous recombination-independent approach was established, and mouse adiponectin-UCP1 was efficiently inserted into the porcine endogenous UCP1 locus. The resultant UCP1 KI pigs showed an improved ability to maintain body temperature, decrease fat deposition, and increase carcass lean percentage. UCP1 KI pigs are a potentially valuable resource for the pig industry that can improve pig welfare and reduce economic losses [15]. Due to the critical role of beige adipocytes in fighting against obesity in humans, the appearance of beige adipocytes in certain pig breeds has sparked debate that the induction of beige in pigs may provide an alternative way to decrease fat deposition, therefore improving the production efficiency in pigs.

However, in our previous study, beige adipogenesis was identified in acute cold-treated adipose tissues from cold-tolerant pig breeds, such as Tibetan pigs [16]. Our group successfully established porcine white and beige adipocytes from the SVFs of Tibetan pigs in in vitro differentiation systems and characterized both cells at the cellular and molecular levels [17]. Significantly, except for the well-recognized BAT/beige adipocyte markers in humans and rodents, such as EBF2 [18], TNF receptor superfamily member 9 (CD137) [10,11], mitochondrial transcription factor A (TFAM) [19,20], PPARG coactivator 1 alpha (PPARGC1A) [21], type 2 iodothyronine deiodinase (DIO2) [22,23], and potassium channel K3 (KCNK3) [24], we also identified several induced key transcriptional factors in pigs, including bone morphogenetic protein 2 (BMP2), suggesting its pivotal role in beige adipogenesis. BMPs belong to the transforming growth factor-β (TGF-β) superfamily, which includes more than 20 members [25]. BMP4 and BMP7 have been identified as the critical transcriptional factors to determine brown adipocyte fate [26], whereas BMP6 has been implicated in inducing brown fat differentiation within skeletal muscle precursor cells [27]. Despite the fact that BMP2 and its closest relative, BMP4, have been reported to be required for the commitment of C3H10T1/2 [28,29], their role in pig beige adipogenesis remains unclear.

Here, we endeavor to investigate the essential role of BMP2 expression in porcine beige adipocyte differentiation and explore the detailed molecular mechanisms. Our study holds the potential to improve meat production in pigs but also holds promise for advancing human obesity research.

## 2. Results

### 2.1. BMP2 Expression Was Highly Induced in the Early Stages of Differentiation

To investigate the potential involvement of *BMP2* in porcine beige adipocyte differentiation, porcine SVFs from Tibetan pigs were primarily isolated, cultured and differentiated into beige adipocytes, as previously described [17]. Figure 1A shows that schematic ORO staining, qPCR, and Western blotting were used to assess the extent of beige adipogenic differentiation, and, as shown in our results, the intensity of ORO staining exhibited a gradual increase from day 0 to day 8 (Figure 1B,C). Consistent with this result, the pivotal adipogenic markers (*PPARγ*, *C/EBPα*, and *FABP4*) (Figure 1D), as well as beige adipocyte-specific genes such as *CIDEA*, *CD137*, *P2RX5*, *PGC1A*, *CITED1*, *EBF2*, and *UCP3* (Figure 1E,F), exhibited a progressive upregulation over the course of the 8-day adipogenic differentiation process at mRNA levels. Some of them, such as PPARγ, CEBPɑ, PRDM16, were further detected at protein level (Figure 1H). Moreover, *BMP2* expression experienced a substantial upregulation as early as the second day of differentiation and maintained a high-expression level during the differentiation (Figure 1G,H), suggesting its critical role in beige adipocyte differentiation.

### 2.2. BMP2 Regulates Porcine Beige Adipogenesis at the Early Stage of Differentiation

To demonstrate the role *BMP2* plays during the initial stages of porcine beige adipogenesis, porcine SVFs were treated with various concentrations of purified recombinant human BMP2 (rhBMP2) from −2 days to 2 days of differentiation (Figure 2A). After eight days of induction, we assessed porcine beige adipocyte differentiation by Oil Red O staining and BODIPY fluorescence staining. Compared to the control group without rhBMP2, the rhBMP2 treatments led to a more pronounced lipid accumulation, with the degree of accumulation increasing with higher rhBMP2 concentrations (Figure 2B–D). Consistently, adipogenic genes, including *PPARγ*, *C/EBPɑ*, *FABP4*, and *FASN* etc., and thermogenic and beige marker genes, such as *CIDEA*, *DIO2*, *CD137*, *ELOVL6*, and *PGC1α*, all displayed a dose-dependent increase in their response to rhBMP2 treatment at both RNA and protein levels (Figure 2E–H). Furthermore, SVFs treated with 50 ng/mL of rhBMP2 exhibited significantly higher basal respiration rates compared to those of untreated SVFs (Figure 2I–K), which probably originated from an increased number of mitochondria due to efficient differentiation (Figure 2L). Notably, treatments with rhBMP2 during the intervals of 2–4 days, 4–6 days, and 6–8 days (Appendix A) did not significantly alter the efficiency of porcine beige adipocyte differentiation in SVFs (Appendix A), together with the unchanged gene expression levels (Appendix A–H), illustrating its role in the early stages of differentiation.

Furthermore, porcine SVF cells were transfected with si*BMP2* or negative control (siNC) from −2 d to 2 d during beige adipogenesis (Appendix A), and the decreased expression of *BMP2* was confirmed (Appendix A). The compromised beige adipocyte differentiation was observed in si*BMP2*-transfected cells with the evidence of Oil Red O staining (Appendix A) and the decreased mRNA expression levels of adipogenic and thermogenic genes (Appendix A). Collectively, these findings emphasize the significant role of BMP2 during the early stages of porcine beige adipogenesis.

### 2.3. BMP2 Enhances Porcine SVFs Proliferation

One of the earliest events in the process of adipogenesis involves a brief proliferative phase, which is subsequently followed by cell-cycle arrest and terminal differentiation [30]. To investigate the impact of *BMP2* during the initial stages of porcine beige adipogenesis, we assessed the influence of rhBMP2 on the proliferation of porcine SVFs. We examined the proportion of 5-ethynyl-2′-deoxyuridine (EdU)-positive cells among porcine SVFs following rhBMP2 treatment. Representative images of EdU staining (Figure 3A) and quantitative analysis (Figure 3B) revealed that pretreatment with rhBMP2 resulted in a substantial increase in cell proliferation 40 h after the initiation of adipogenesis. This effect was further supported by the upregulation of the cell proliferation marker, proliferating the cell nuclear antigen (PCNA) at both day 0, day 1, and day 2 in cells subjected to rhBMP2 pretreatment (Figure 3C).

Numerous studies reported that early adipocyte differentiation is characterized by the induction of at least two families of transcription factors, C/EBP and PPAR, with sterol regulatory element-binding protein 1c (SREBP-1C) also being induced during the early stages of adipocyte differentiation. Consistent with these findings, on day 0, day 1, and day 2 of induction, we observed a significant upregulation in the expression of the following three early differentiation transcription factors, C/EBPα, SREBP-1C, and PPARγ, in rhBMP2-treated cells (Figure 3C–F). Additionally, considering that the phosphorylation and translocation of C/EBPβ play crucial roles in activating adipogenic transcription factors during the initial stages of adipogenesis, we evaluated the levels of phosphorylated C/EBPβ (pC/EBPβ) in both cell groups. The levels of pC/EBPβ were observed to be increased in rhBMP2-pretreated cells (Figure 3E,G).

### 2.4. BMP2 Positively Regulates the Porcine Beige Adipogenic Differentiation and Thermogenesis In Vitro

To explore the mechanism of beige adipogenesis mediated by *BMP2*, we constructed overexpression and knockdown plasmids, *BMP2*-OE and *BMP2*-sh, respectively, which were transfected into porcine SVF cells and differentiated into the beige adipocytes (Figure 4A). The transduction efficiency of these plasmids was notably high (Appendix A), and ORO staining unequivocally demonstrated that *BMP2* overexpression led to a remarkable augmentation in the formation of beige adipocytes, while *BMP2* knockdown decreased the beige adipogenesis compared to the empty plasmid transfection control (Figure 4B). As expected, *BMP2* was highly increased in *BMP2*-OE-transfected cells and significantly decreased in *BMP2*-sh-transfected cells (Figure 4C). Consistently, the expression levels of adipogenic genes, such as C/EBPɑ, FASN, ACC, beige adipocyte marker genes, including CIDEA and PRDM16, and mitochondrial genes, such as ATP5A, NDUFB8, and SDHB, were greatly increased in *BMP2* overexpressed adipocytes, while they were slightly decreased in *BMP2* knockdown adipocytes (Figure 4D and Appendix A). Furthermore, a Seahorse assay was used to evaluate the effects of *BMP2* on uncoupled mitochondrial respiration and fatty acid oxidation in primary porcine beige adipocytes. Our data demonstrated that *BMP2* overexpression significantly enhanced the basal mitochondrial respiration rate (Figure 4E,F), maximal respiration (Figure 4G), and proton leak (Figure 4H), while the knockdown of *BMP2* led to the opposite effects (Figure 4E–H). Above all, these results affirm that *BMP2* exerts a positive regulatory influence on the adipogenic and thermogenic capacity of porcine beige adipocytes in vitro.

To explore the key signaling pathways that are involved in *BMP2*-mediated beige adipogenesis, RNA-seq analysis was conducted for three groups of cells. Principal component analysis (PCA) clearly distinguished the three experimental groups, indicating the distinct molecular networks among groups (Appendix A). Volcano plots were built based on the criteria of *p*-value and fold change, and the differentially expressed genes (DGEs) were screened under the criteria of *p* < 0.05 and fold change > 1. We are particularly interested in the signaling pathways that were significantly affected in both *BMP2*-overexpressed and *BMP2* knockdown cells. Thus, a total of 1273 significantly upregulated genes in the *BMP2*-OE group (Appendix A) were merged with 903 significantly downregulated genes from *BMP2*-sh cells (Appendix A) and 213 genes were identified (Appendix A, Figure 4I). These genes encompassed key adipogenic genes such as *C/EBPα*, *ADIPOQ*, *SCD*, *FASN*, and *FABP4*, along with genes associated with thermogenesis and beige adipocytes, including *UCP3*, *VEGFA*, and *TMEM26* (Figure 4J). The gene ontology (GO) analysis of 213 DEGs unveiled that GO annotations of brown fat cell differentiation, the lipid biosynthetic process, cold-induced thermogenesis, and adaptive thermogenesis were enriched. In addition, a significant enhancement of the PPAR signaling pathway was found in KEGG analysis, indicating its potential role in beige adipogenesis and thermogenesis (Figure 4L). Additionally, the analysis also highlighted the significant enrichment of cell adhesion, ERK1 and ERK2 proliferation, mesenchymal cell proliferation, and the MAPK cascade (Figure 4K).

### 2.5. BMP2 Promotes Adipogenesis via PPARγ

As described above, *BMP2* affected the expression of *PPARγ*, a core transcriptional factor that is involved in adipogenesis; to further corroborate the central role of *BMP2*-mediated *PPARγ* regulation in porcine beige adipogenesis, the PPARγ antagonist T0070907 was supplemented (Figure 5A) and its effect on lipogenesis was examined. We designed four experimental groups. In the first group, neither rhBMP2 nor T0070907 was added during the differentiation process (rhBMP2−T0070907−); the second group was given supplemented T0070907 at a concentration of 10 μM without rhBMP2 (rhBMP2-T0070907+); the third group was treated with 50 ng/mL of rhBMP2 without T0070907 (rhBMP2+T0070907−); and the fourth group was supplemented with both 50 ng/mL of rhBMP2 and 10 μM of T0070907 (rhBMP2+T0070907+). Our results indicate that treatment with T0070907 significantly reduced the expression levels of PPARγ (Figure 5D) and markedly decreased the differentiation of porcine beige adipocytes (Figure 5B). Compared to rhBMP2+T0070907−, rhBMP2+T0070907+ showed a highly significant reduction in the differentiation efficiency of porcine beige adipocytes (*p* < 0.001), suggesting that the addition of T0070907 diminished the effect of rhBMP2 in promoting the differentiation of porcine beige adipocytes (Figure 5B,C). Consistently, the expression levels of adipogenic-related genes, including *FABP4*, *FASN*, *ADIPOQ*, *ACACA*, *SCD*, and *AP2* (Figure 5D,E), and beige/brown adipocyte markers, including *PGC1ɑ*, *EBF2*, *DIO2*, and *SYNE2* (Figure 5F), were greatly decreased in T0070907-treated cells and the T0070907 supplement, also suppressed the *BMP2*-mediated induction of gene expression at the RNA level. Some of them, such as *PPARγ*, *ACACA*, *FASN*, *CEBPɑ*, *PGC1α*, and *PRDM16*, were further confirmed at the protein level (Figure 5G). These data suggest that the promoting effects of rhBMP2 on adipogenesis and thermogenesis are abolished following the silencing of PPARγ, highlighting the essential role of PPAR*γ* in mediating BMP2’s modulation of porcine beige adipocyte adipogenesis.

### 2.6. BMP2 Upregulates the Transcriptional Activity of PPARγ through MAPK and AKT/mTOR Signaling

Extensive studies revealed that the expression of *PPARγ* was regulated by the MAPK pathway [31]; we also examined whether the MAPK signaling pathway was affected in rhBMP2-treated cells at the onset of differentiation (Figure 6A). Our data showed that the expression levels of the genes related to the Raf-MEK1/2-ERK1/2 signaling pathway, including *RAF1*, *MEK1*, *MEK2*, *ERK1*, and *ERK2* (Figure 6B–D), and the genes related to the p38 signaling pathway, including *MKK6* and *MKK3* (Figure 6G), were significantly increased at the early stage of differentiation. Furthermore, a significant increase in the phosphorylation and total protein levels of ERK1/2 (Figure 6E,F) and P38 (Figure 6H,I) was found in rhBMP2-treated cells.

Considering that the adipogenic cocktail-induced activation of the AKT/mTOR/S6 axis plays an essential role in cell proliferation and metabolism during adipogenesis and lipogenesis [32], we investigated whether the AKT/mTOR/S6 axis was also involved in *BMP2*-mediated adipogenic effects. Interestingly, our results demonstrated that the addition of rhBMP2 promoted the phosphorylation of AKT and S6K (Figure 6J). Furthermore, *BMP2*-overexpressed cells exhibited induced protein levels of phospho-AKT and phospho-S6 (Appendix A), compared to the control group, while *BMP2*-deficient cells exhibited reduced protein levels. These results suggest that *BMP2* activates the MAPK and AKT/mTOR/S6 signaling pathways.

Moreover, we also examined the roles of the MAPK and mTOR pathways in the regulation of the PPARγ function. We found that a supplement of the PPARγ inhibitor, T0070907, abolished the rhBMP2-induced ERK1/2, p38, AKT, and S6K phosphorylation (Figure 6K). Taken together, *BMP2* may influence the activity of PPARγ through the MAPK and AKT/mTOR signaling pathways, thereby affecting early beige adipogenesis (Figure 7).

## 3. Discussion

Here, we found that *BMP2* regulates the porcine beige adipogenesis and thermogenesis positively. Like brown adipocytes, beige adipocytes dissipate energy and are considered a possible therapeutic intervention for obesity and its related metabolic diseases [33]. At this point, *BMP2* may serve as the potential candidate gene for lower fat deposition in pigs. The research on *BMP2* in pigs was majorly focused on bone morphogenesis and development. For example, the study of the genome-wide association analysis (GWAS) in a commercial Duroc × (Landrace × Yorkshire) pig population revealed that the rs320706814 SNP, located approximately 123 kb upstream of the *BMP2*, was the strongest candidate causal mutation for carcass length [34]. The association of *BMP2* and carcass length was also observed in other populations, such as Piétrain, Large White, Landrace, and Meishan [35]. In addition, the SNPs, rs321846600 and rs1111440035, of *BMP2* were identified as candidate SNPs that are functionally related to the Yorkshire pigs’ loin muscle depth [36].

Despite the fact that our study illustrated the novel role of *BMP2* in porcine adipogenesis, its potential implication in the improvement of fat-related traits still needs to be further explored. On one hand, the SNPs that could increase the expression of *BMP2* need to be screened in various pig populations, and the association studies between such SNPs and fat-related traits, such as backfat thickness and fat content, should be performed. If such SNPs can be identified, they might be used for lower fat deposition in the pig genetic selection program. On the other hand, the rapid development of genome editing techniques makes it possible to produce genetically modified pigs for the dramatic improvement of fat-related traits. For example, it has been reported that the adipocyte-specific knockdown of the Ucp1 gene in pigs significantly decreases the backfat thickness and fat content and improves the body temperature’s regulatory capacity [15]. Thus, we propose that the overexpression of the *BMP2* gene in fat tissues enhances thermogenesis and decreases the fat content in pigs. This study is required to address its in vivo role in fat deposition in pigs.

Three members of the BMP family have been demonstrated to play critical roles in adipocyte commitment and differentiation. *BMP7* has been well-recognized to be sufficient to induce brown adipocyte commitment [26], while *BMP4*, secreted by white adipocytes, preferentially regulates the beige/brown phenotype and serves as an integral feedback regulator of both white and beige adipogenic commitment and differentiation [37]. In addition, *BMP2* was proven to be involved in the differentiation of white adipocytes [31] and C3H10T1/2 cells, which is a cell model of brown adipocytes. In our previous study, we characterized the molecular features of porcine beige adipocytes and identified *BMP2*, but not *BMP4* and *BMP7*, as a candidate gene for the regulation of beige formation [17]. Here, we provide robust evidence that *BMP2* positively regulates porcine beige adipogenesis and thermogenesis. However, whether *BMP2* exerts a critical role in beige adipocyte commitment in pigs still needs to be further explored by genetic tracer systems.

Consistent with the observations in C3H10T1/2 cells [38], our data demonstrated that *BMP2* activates the ERK, p38, and AKT signaling pathways and contributes to the transcriptional activation of *PPARγ* and *C/EBPα*, resulting in the promotion of adipogenesis [39,40]. However, the detailed effects of these pathways in *BMP2*-induced *PPARγ* need to be further explored by specific inhibitors.

In summary, our study provides compelling evidence for the pivotal role of *BMP2* in the adipogenic and thermogenic capacities of porcine beige adipocytes and enhances our understanding of porcine beige adipogenesis. More importantly, we identify *BMP2* as a promising candidate gene for reducing fat deposition in pigs. Beyond its significance in agriculture, our study also provides valuable data for cross-species studies in the field of metabolism, especially obesity and its related metabolic diseases in humans.

## 4. Materials and Methods

### 4.1. Compliance with Ethical Requirements

All Institutional and National Guidelines for the care and use of animals were followed. This study was reviewed and approved by the Animal Ethics Committee of the Institute of Animal Science, Chinese Academy of Agricultural Sciences (31 March 2020, CAAS; approval No: IAS2020-21).

### 4.2. Isolation, Culture, and Differentiation of Porcine SVF Cells

The isolation, culture and differentiation processes of porcine SVF cells were described previously [17]. Briefly, adipose tissues were collected from three 1-month-old male Tibetan piglets. Adipose tissue was dissected, washed, digested, filtered, and centrifuged to pellet the SVF cells. For beige adipogenic differentiation, SVFs were grown for an additional 2 days after reaching 100% confluence and then induced to differentiate with high-glucose DMEM supplemented with 1% P/S, 2% FBS, 1 nM of T3, 1 μM of Dexamethasone, 0.5 mM of isobutylmethylxanthine and 2 μM of Rosiglitazone and the medium was replaced every other day for 8 days.

### 4.3. Western Blot

Beige adipocytes were lysed with the M-PERTM Mammalian Protein Extraction Reagent (#78503, Thermo Fisher Scientific, Waltham, MA, USA) supplemented with a cocktail of protease inhibitors (#04693159001, Roche, Indianapolis, IN, USA). The cell lysate was centrifuged at 12,000× *g* for 20 min, and the supernatant was used for the following study. Proteins (20–50 μg) were separated by 10% SDS-PAGE and then transferred onto PVDF membranes (IPVH00010, Merck-Millipore, Madison, WI, USA). The membranes were sealed with 5% skimmed milk/TBST for 2 h and then incubated with primary antibodies at 4 °C overnight. The membranes were rinsed with TBST three times for 10 min each and then incubated with secondary antibodies. The protein bands were captured by a FluorChem M Fluorescent Imaging System (Tanon 5200, Tanon, Shanghai, China) with a TanonTM High-sig ECL Western blotting substrate (#180501, Tanon, Shanghai, China). The antibodies are listed in Appendix A.

### 4.4. RNA Isolation, RT-PCR, and qPCR Analysis

The cells were washed with DPBS and lysed directly in 12-well plates with the TRIzol reagent (#15596018, Thermo Fisher Scientific, Waltham, MA, USA). Total RNA was extracted according to the manufacturer’s instructions. The RNA was reverse-transcribed using the PrimeScriptTM RT Reagent Kit with a gDNA Eraser (RR047A, Takara, Tokyo, Japan). RT-qPCRs were run using TB Green^®^ Premix Ex TaqTM (RR420A, Takara, Tokyo, Japan). Target gene expression levels were normalized to 18S expression. Primer sequences are listed in Appendix A.

### 4.5. mRNA Sequencing, RNA-Seq Data Analysis and Functional Analysis

Sequencing libraries and RNA-seq were conducted at Shanghai Personal Biotechnology Co., Ltd. (Shanghai, China) RNA samples with high purity (OD 260/280 ≥ 2.0) and high integrity (RIN > 7) were used to construct the cDNA library. More detailed information is presented in a previous study [16]. Genes with a fold change of (FC) > 1.0 and *p* < 0.05 after correcting for multiple testing were classified as differentially expressed genes (DEGs). The gene ontology (GO) enrichment analysis of upregulated and downregulated genes was performed using the Metascape database (https://metascape.org, accessed on 16 June 2021). GO terms with *p* < 0.05 were regarded as statistically significant. R studio, based on R version 4.1.0, and an online website was used for statistical analysis and mapping (https://software.broadinstitute.org/morpheus, accessed on 16 June 2021).

### 4.6. Oil Red O Staining

Cells were carefully washed twice with DPBS and fixed with 4% paraformaldehyde at room temperature for 30 min to 1 h. After washing with DPBS twice, the cells were stained with a 60% saturated Oil Red O reagent (G1260, Solarbio, Beijing, China) for 15 min and then washed with 60% isopropanol and DPBS. The Oil Red O absorbed by cells was extracted with 100% isopropanol and detected by a SpectraMax M5 (Molecular Devices, San Jose, CA, USA) at 510 nm.

### 4.7. Fluorescence Microscopy

Cells were seeded, cultured and differentiated in glass-bottom confocal plates (D35-20-0-N, Cellvis, Hangzhou, China). On the day of the experiment, 200 nM of BODIPY493/503 (D3922, Thermo Fisher Scientific) was added to the cell culture medium for 30 min, followed by imaging, which was performed on a laser scanning confocal microscope (Leica TCS SP8, Wetzlar, Germany). A 63 × apochromatic oil-immersion mirror and AiryScan super-resolution detector were used for super-resolution imaging. The low-resolution images were taken with a 10 × air objective lens. All fluorophores were excited in separate orbits to avoid artifacts due to bleed-through emission. BODIPY 493/503 was excited at 488 nm.

### 4.8. EdU Cell Proliferation Assay and Analysis

5′-ethynyl-2′-deoxyuridine (EdU) incorporation was assessed using the Cell-Light EdU Apollo567 In Vitro Kit (Guangzhou RiboBio, Guangzhou, China) according to the manufacturer’s recommendation. Briefly, porcine SVFs were allowed to reach confluency (referred to as day −2) in a 48-well cell imaging plate (Corning, NY, USA) and maintained at confluency for 2 days (referred to as day 0). Differentiation was initiated at day 0, and EdU was added at a final concentration of 5 μM after 16 h of differentiation. After a total of 40 h of differentiation, the cells were fixed in 4% PAF for 30 min at room temperature and permeabilized with 0.5% Triton X-100. Subsequently, a 1 × Apollo reaction cocktail was added to the cells and incubated for 30 min, and then the cells were stained with Hoechst 33342 for DNA content analysis. Finally, EdU-positive cells (EdU^+^) were visualized under a fluorescence microscope (Nikon, Tokyo, Japan). The analysis of porcine SVF proliferation (ratio of EdU^+^ cells to all cells) was performed using images of randomly selected fields (*n* = 9) obtained under a fluorescence microscope.

### 4.9. Seahorse Metabolic Assays

SVFs (1 × 10^4^) were inoculated into an XFe96 culture microplate (101085-004, Seahorse Bio-science, North Billerica, MA, USA)) and cultured in DMEM containing 10% FBS and 1% P/S at 37 °C in a 5% CO_2_ atmosphere, followed by the induction of differentiation. On day 8, O_2_ consumption was measured using a Seahorse Bioscience XFe96 extracellular flux analyzer. The oxygen consumption rate (OCR) was determined by sequentially adding 1.5 M of oligomycin, 0.5 μM of carbonyl cyanide p-(trifluoromethoxy) phenylhydrazone (FCCP) and 0.5 of μM antimycin A/Rotenone. The basal respiration rate, proton leak and ATP production were calculated by Wave Desktop webpage (https://seahorseanalytics.agilent.com/) (Agilent, Palo Alto, CA, USA). All data are shown as the mean ± standard error mean (SEM). Statistical comparisons were made using Student’s *t*-test.

### 4.10. RNA Interference

Stealth RNAi^TM^ against pig BMP2 (si-BMP2) (Jima, Shanghai, China) was used to perform RNA interference. SVF cells cultured in 12-well plates and grown to 80% confluence were transfected with siRNA with Lipofectamine RNAiMAX (#13778150, Invitrogen, Waltham, MA, USA) according to the manufacturer’s instructions. All transfection experiments were performed in triplicate. The siRNA sequences were as follows: negative control (si-NC): 5′-UUCUCCGAACGUGUCACGUTT-3′; antisense: 5′-ACGUGACACGUUCGGAGAATT-3′; si-BMP2: 5′-GACCCUUGCUAGUCACUUUTT-3′; and antisense: 5′-AAAGUGACUAGCAAGGGUCTT-3′.

### 4.11. Statistical Analysis

All experiments were performed with at least three biological replicates unless otherwise noted. For statistical analysis, GraphPad Prism 8.0.2 and JMP 10.0.0 were used. Data that were normally distributed were analyzed using Student’s two-tailed and unpaired *t*-test. All values are expressed as the mean ± SEM. * *p* < 0.05, ** *p* < 0.01 and *** *p* < 0.001 were used as the statistically significant standards.

## Figures and Tables

**Figure 1 ijms-25-03915-f001:**
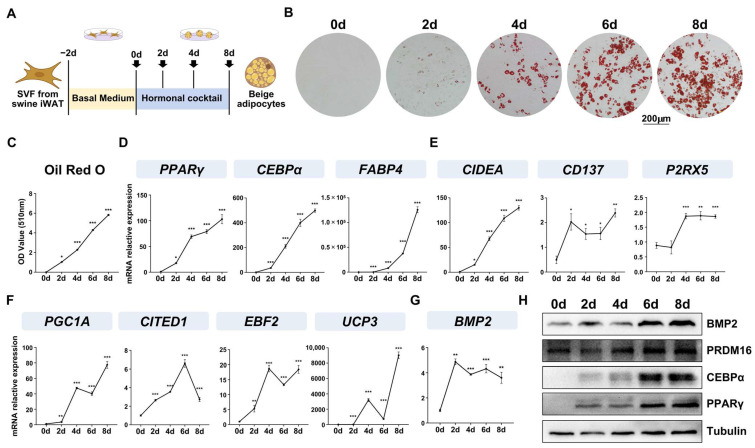
*BMP2* expression was highly induced in the early stages of differentiation (**A**) Schematic representation of the experimental design. (**B**) ORO staining gradually increased from 0 to 8 days (scale bar: 200 μm); (**C**) Quantitative analysis of the ORO staining data shown in B via OD measurements, *n* = 6; qPCR showed that the mRNA levels of adipogenic (**D**) and thermogenic genes, and beige adipocytes markers (**E**,**F**) and *BMP2* (**G**) were increased during adipogenic differentiation, *n* = 4; data are presented as the mean ± SEM, * *p* < 0.05, ** *p* < 0.01, *** *p* < 0.001; (**H**) Western blotting showed that the protein levels of *PPARγ*, *C/EBPα* and *PRDM16* were gradually increased during adipogenic differentiation, while *BMP2* increased after induction toward adipogenic differentiation.

**Figure 2 ijms-25-03915-f002:**
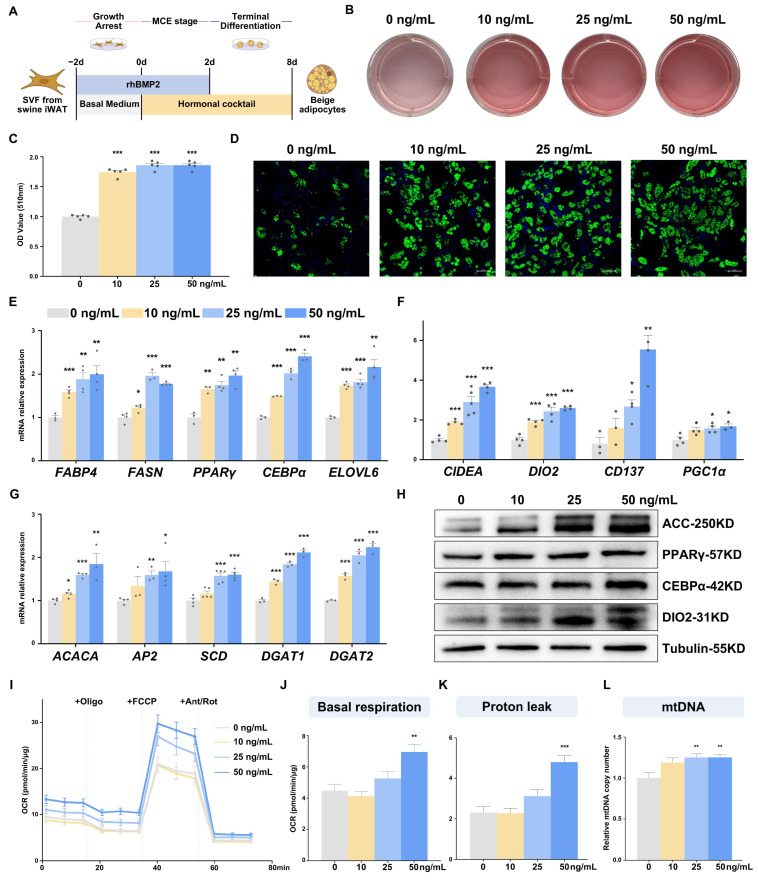
Pretreatment with rhBMP2 in the early stages of differentiation enhanced porcine SVFs’ adipogenic and thermogenic capacity (**A**) Schematic representation of the experimental design. (**B**) Adipogenesis assessed by Oil Red O staining on day 8 of differentiation. (**C**) Quantitative analysis of Oil Red-O staining data show in B via OD measurements that *n* = 5. (**D**) The fluorescence staining of beige adipocytes (the scale bar is 50 μm), lipid droplets (BODIPY 493/503, green), nucleus (DAPI, blue). Adipogenesis-related gene (**E,G**) and beige adipocyte-related gene (**F**) expression via qPCR at 8 days of differentiation, *n* = 4. (**H**) Western blot analysis of the expression of adipogenic and thermogenic marker genes. (**I**–**K**) OCR in differentiated porcine beige adipocytes with rhBMP2 pretreatment. The basal cellular respiration rate (**J**) and proton leak (**K**) were calculated. Seahorse assay data were normalized to the total protein concentration, *n* = 10. (**L**) The relative mtDNA copy number is *n* = 6. All data are presented as the mean ± SEM, * *p* < 0.05, ** *p* < 0.01, *** *p* < 0.001.

**Figure 3 ijms-25-03915-f003:**
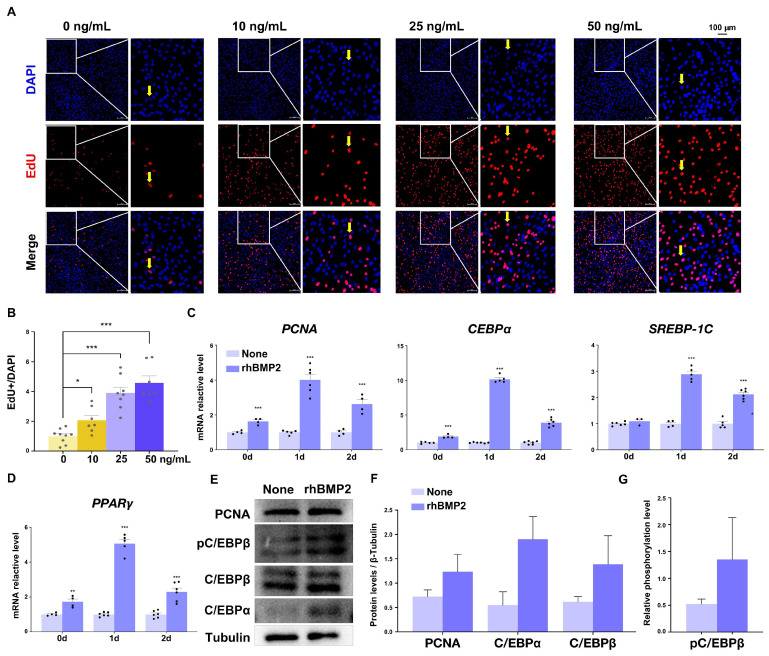
Pretreatment with rhBMP2 enhanced porcine SVF proliferation. (**A**) The EdU assay was used to analyze cell proliferation in rhBMP2-treated cells. EdU was added at a final concentration of 5 μM after 16 h of differentiation. (**B**) Quantitative data from (**A**) where *n* = 9. (**C**) The mRNA expression level of proliferation marker *PCNA*, adipogenic transcription factors *C/EBPα*, *SREBP-1C*, and *PPARγ* (**D**), in non-treated and rhBMP2-treated cells was detected by qPCR and *18S* was used as an internal control, where *n* = 6. (**E**) The expression levels of proteins, including proliferation marker PCNA, early adipogenesis markers (C/EBPβ and p-C/EBPβ), and adipogenic transcription factors (C/EBPα) in non-treated and rhBMP2-treated cells. β-Tubulin was used as a loading control. (**F**,**G**) Data quantification of panel (**E**), *n* = 3. All data are presented as the mean ± SEM, * *p* < 0.05, ** *p* < 0.01, *** *p* < 0.001.

**Figure 4 ijms-25-03915-f004:**
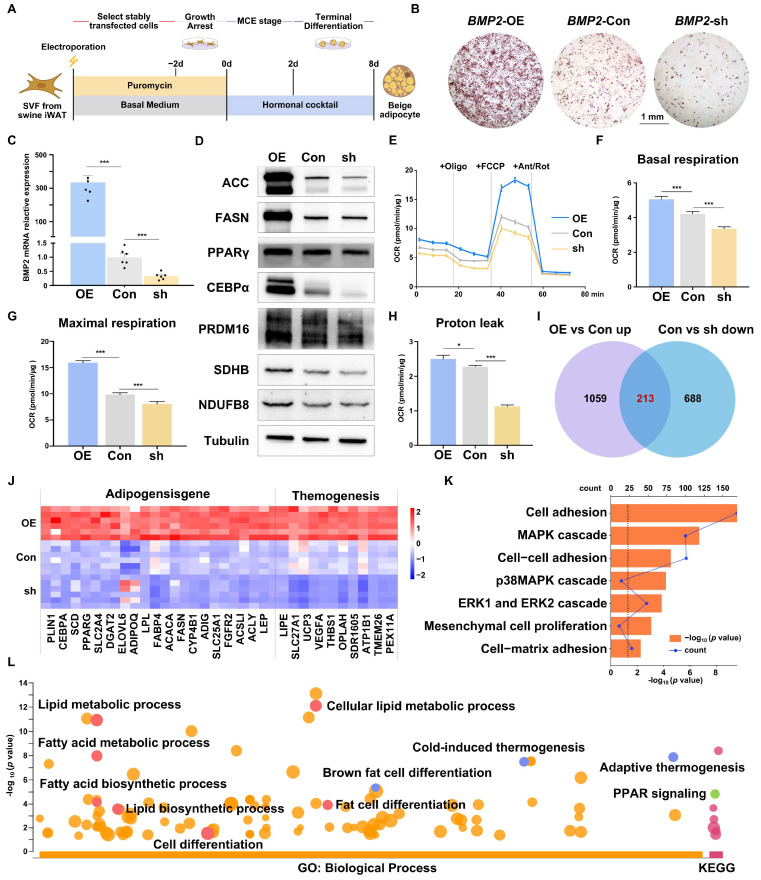
Overexpressed *BMP2* enhances lipid accumulation and thermogenesis, while knockdown *BMP2* has the opposite effect. (**A**) Schematic representation of the experimental design. (**B**) Adipogenesis assessed by Oil Red O staining at day 8 of differentiation. (**C**) Overexpression and ablation efficiency of *BMP2* in porcine SVFs, *n* = 6. (**D**) Western blot analysis of the expression of adipogenic and thermogenic marker genes. (**E**–**H**) OCR in differentiated porcine beige adipocytes with *BMP2* overexpression or knockdown (**E**). The basal cellular respiration rate (**F**), maximal respiration (**G**), and proton leak (**H**) were calculated. The seahorse assay data were normalized to the total protein content, *n* = 10. All data are presented as the mean ± SEM, * *p* < 0.05, *** *p* < 0.001. (**I**) Venn diagram based on the number of DEGs. (**J**) Heatmap and detailed list of genes with the same expression trends. (**K**) The pathways associated with early adipogenesis (bar: −log_10_ *p*-value; line: gene counts, dashed line: *p* < 0.05). (**L**) GO analysis and KEGG enrichment of 213 genes overlapping in (**I**). (Blue: thermogenic pathway; Red: adipogenic pathway; Green: PPAR signaling; Orange: other pathway in GO analysis; Rose red: other pathway in KEGG).

**Figure 5 ijms-25-03915-f005:**
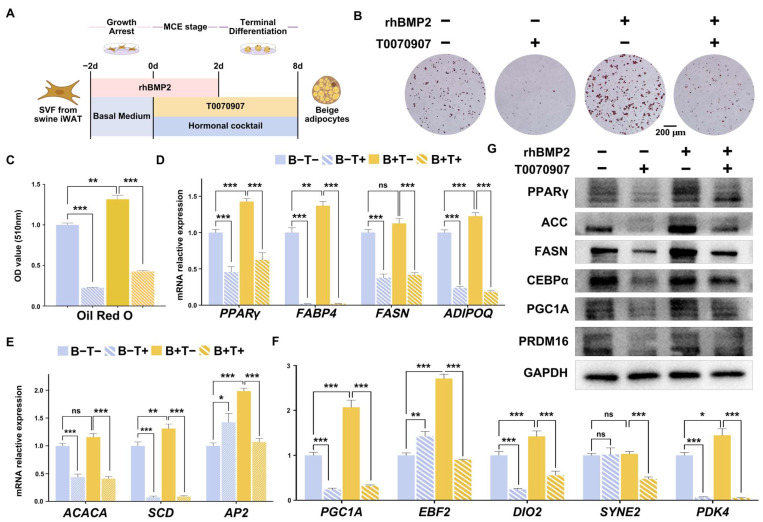
rhBMP2 treatment promoted cellular lipogenesis via PPARγ. (**A**) Schematic representation of the experimental design. (**B**) Adipogenesis assessed by Oil Red O staining on day 8 of differentiation. (**C**) Quantitative analysis of the Oil Red O staining data show. in B via OD measurements, *n* = 4. (**D**,**E**) Adipogenesis-related genes and (**F**) beige adipocyte-related gene expression by qPCR at 8 days of differentiation, *n* = 9. (**G**) Western blot analysis of the expression of adipogenic and thermogenic marker genes. All data are presented as the mean ± SEM, ns *p* > 0.05, * *p* < 0.05, ** *p* < 0.01, *** *p* < 0.001.

**Figure 6 ijms-25-03915-f006:**
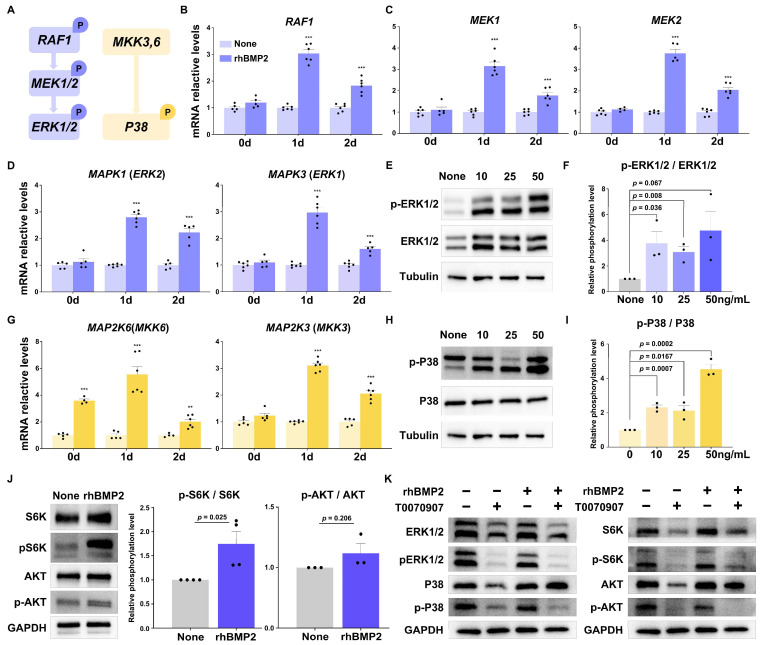
*BMP2* upregulates the transcriptional activity of PPARγ through MAPK and AKT/mTOR signaling. (**A**) Schematic diagram of RAF-MEK1/2-ERK1/2 and MKK3,6-P38 signaling pathway. (**B**–**D**) Expression levels of the genes involved in the Raf-MEK1/2-ERK1/2 signaling pathway. 18S was used as an internal control, *n* = 6. (**E**) Representative Western blot shows the total protein and phosphorylation levels of ERK1/2 on day 2 after induction. β-Tubulin was used as a loading control. (**F**) Data quantification of panel (**E**), where *n* = 3. (**G**) Expression levels of the genes involved in the MKK3,6-P38 signaling pathway, where *n* = 6. (**H**) Representative Western blot showing the total protein and phosphorylation levels of P38 on day 2 after induction. (**I**) Data quantification of panel (**H**), where *n* = 3. (**J**) Representative Western blot and quantification of the panel showing the total protein and phosphorylation levels of AKT and S6K in rhBMP2-treated cells, where *n* = 4. All data are presented as the mean ± SEM, ** *p* < 0.01, *** *p* < 0.001. (**K**) Western blot showing the total protein and phosphorylation levels in rhBMP2- and T0070907-treated cells.

**Figure 7 ijms-25-03915-f007:**
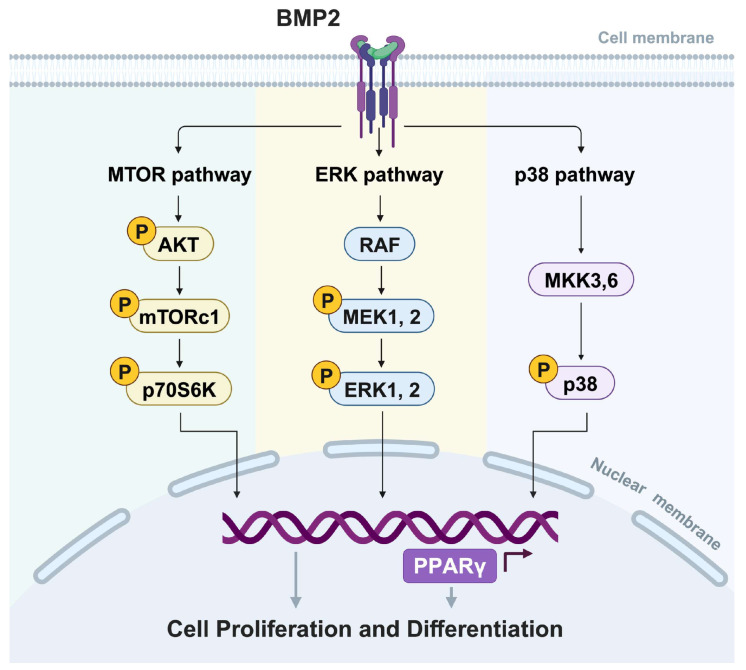
*BMP2* promoted cellular lipogenesis via PPARγ through the AKT/mTOR and MAPK signaling pathways.

## Data Availability

The RNA-seq raw data are available in the Genome Sequence Archive in BIG Data Center, Beijing Institute of Genomics (BIG), Chinese Academy of Sciences, under accession number CRA015105 (http://bigd.big.ac.cn/gsa, Public) (accessed on 27 February 2024).

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
