# Peer review of "Bone Morphogenetic Protein 2 Enhances Porcine Beige Adipogenesis via AKT/mTOR and MAPK Signaling Pathways"

_ijms, 2024, doi:10.3390/ijms25073915_

Round 1
Reviewer 1 Report
Comments and Suggestions for Authors The heat production of pig adipose tissue is a very interesting topic. Liu et al.'s paper investigated the role of BMP2 in SVF cells and discussed its potential role in the browning of pig adipocytes. The overall idea of the manuscript is clear and the writing is fluent, but there are some issues that need improvement or supplementation. 1. This manuscript uses SVF cells derived from Tibetan pigs. Tibetan pig is a small pig breed unique to the plateau, which has strong cold resistance. Is there any difference in the adipose tissue between Tibetan pigs and ordinary pigs?, Is the selection of SVF cells from Tibetan pigs representative? Therefore, it is recommended to partially supplement the experiments on SVF cells in ordinary pigs. 2. The manuscript focuses on studying the beige staining of adipose tissue, but too few molecular markers for beige staining of adipose tissue are used (tbx1, tmem26, fgf21, p2rx5, pat2, car4, etc. should be supplemented), and there is a lack of phenotype data for cell beige staining (such as electron microscopy photos). 3. All bar charts have no annotated sample size, and the WB results do not show biological replicates. 4. This manuscript extensively utilizes rhBMP2, but does not explain the homology and similarity of BMP2 between humans and pigs. Human beings have brown fat and pigs lack brown fat, so the suitability of using rhBMP2 needs further consideration. 5. The EDU image and DAPI image in Figure 3B of the manuscript do not seem to overlap and require careful inspection and verification.Author Response
Please see the attachment.

Reviewer 2 Report
Comments and Suggestions for Authors
Authors explained how BMP2 Enhances Porcine Beige Adipogenesis via AKT/mTOR 3 and MAPK Signaling. Manuscript is well written and well structured. I have few concerns though.
Major Comments:
Introductions need to be much clearer. More about Beige adipogenesis, its role and why it could be therapeutic. Why they focus on BMP2. How beige adipocytes involved in meat quality, production etc.
If they will explain the Beige adipocyte isolation from SVF then it will be easier to understand.
Figure1. Mainly explain increased adipogenesis over time. For Beige adipogenesis they only showed CIDEA RT-PCR and PRDM16 Western blot. Though in introduction they mentioned CD137 is a marker of Beige adipocytes. Also, PRDM16 Western blot is not convincing. I would love to see a quantification data.
Figure2. ORO staining result looks same to me for all 3 doses of rhBMP2. Again, this ORO staining and BODIPY 493/503 fluorescence staining showing adipogenesis in early differentiation, not specifically Beige adipogenesis. Why they checked basal respiration? Please explain the correlation and increased mitochondrial number.
Figure3. Quantification of Western Blot data will be convincing.
Figure4. PPARγ WB data is not convincing. It looks GAPDH is not equal. Please use quantification.
BMP2 promotes adipogenesis via PPARγ-Please explain the result section more elaborately.
Comments on the Quality of English LanguageLanguage is fine.
